# CRISPR/Cas9-Mediated Multiplexed Genome Editing in *Aspergillus oryzae*

**DOI:** 10.3390/jof9010109

**Published:** 2023-01-13

**Authors:** Qinghua Li, Jinchang Lu, Guoqiang Zhang, Jingwen Zhou, Jianghua Li, Guocheng Du, Jian Chen

**Affiliations:** 1Science Center for Future Foods, Jiangnan University, 1800 Lihu Road, Wuxi 214122, China; 2National Engineering Research Center for Cereal Fermentation and Food Biomanufacturing, Jiangnan University, 1800 Lihu Road, Wuxi 214122, China; 3School of Biotechnology and Key Laboratory of Industrial Biotechnology, Ministry of Education, Jiangnan University, 1800 Lihu Road, Wuxi 214122, China; 4The Key Laboratory of Carbohydrate Chemistry and Biotechnology, Ministry of Education, Jiangnan University, 1800 Lihu Road, Wuxi 214122, China

**Keywords:** *Aspergillus oryzae*, genetic transformation, gene editing, morphological genes, lipase

## Abstract

*Aspergillus oryzae* has great potential and competitive advantages to be developed as an excellent expression system, owing to its powerful protein secretion ability, complex post-translational modification, and safety characteristics. However, the low efficiency of genetic modification and gene function analysis is an urgent problem to be solved in *A. oryzae* and other filamentous fungal systems. Therefore, establishing efficient genetic transformation and multiplexed genome editing tools is significant for developing *A. oryzae* expression systems, and revealing its intrinsic mechanisms. In this study, the high-efficiency transformation of *A. oryzae* was achieved by optimizing the preparation conditions of protoplasts, and the random editing efficiency of the CRISPR/Cas9 system in *A. oryzae* for single and double genes reached 37.6% and 19.8%, respectively. With the aid of the selection marker, such as color or resistance, the editing efficiency of single and double genes can reach 100%. Based on the developed CRISPR/Cas9 genome editing method, the heterologous lipase gene (*TLL*) achieves precise integration at different genetic loci in one step. The efficient and accurate acquisition of positive transformants indicated that the morphological gene *yA* could be used as a helpful selection marker for genome editing in *A. oryzae*. In conclusion, the developed system improves the efficiency of transformation and multiplexed genome editing for *A. oryzae*. It provides a practical method for developing the *A. oryzae* high-efficiency expression system for heterologous proteins.

## 1. Introduction

*Aspergillus oryzae* is an important filamentous fungus that can use cheap biomass raw materials and is widely used in traditional fermentation and food processing industries [1]. *A. oryzae* has a long history of use in the food industry and is generally recognized as safe (GRAS) by the U.S. Food and Drug Administration (FDA) [2]. Over the past few decades, *A. oryzae* has been developed into an efficient cell factory for organic acids, industrial enzymes, and secondary metabolites due to its strong protein secretion ability and post-translational modification characteristics, and it plays an essential role in the fields of food, medicine, feed, and environment [3,4].

With the in-depth application of *A. oryzae* in many fields, establishing efficient gene editing technology is of great significance for researching gene function and developing cell factories [5]. Spontaneous homologous recombination (HR) has long been the traditional gene-editing method for studying the function of *Aspergillus* genes, but the low efficiency of HR makes the recombination process very cumbersome [6]. Although more efficient HR can be achieved using strains carrying a deletion of genes (*ku70* and *ligD*) involved in non-homologous end joining (NHEJ), deletion of *ku70* or *ligD* by spontaneous HR is more labor-intensive because their multinucleate conidia make it challenging to isolate homokaryotic transformants [7,8].

In recent years, the clustered regularly interspaced short palindromic repeat (CRISPR)-associated Cas9 nuclease has been widely used in the precise gene editing of various microorganisms due to its simplicity, operability, and high efficiency [9,10,11]. However, the editing efficiency of the CRISPR/Cas9 system in *A. oryzae* is relatively low relative to bacteria or yeast. An optimized CRISPR/Cas9 system was constructed and integrated into the genome of *A. oryzae* with an editing efficiency of only 10% to 20% [12]. To this end, researchers have done much work on the efficient expression of Cas9 protein and sgRNA, such as codon optimization of Cas9 protein or expression of related elements using the endogenous promoter of the host strains [13,14]. The CRISPR/Cas9 system based on the episomal expression plasmid pPTR II has been successfully constructed, and its editing efficiency in *A. oryzae* is 50% to 100% [15]. In NHEJ-deficient strains, the editing efficiency of the CRISPR/Cas9 system can also reach nearly 100% using a single-stranded DNA repair template [16]. However, the efficient editing of the CRISPR/Cas9 system is not stable enough for different protospacers of different genes, and requires good screening pressure. Furthermore, efficient HR-based gene editing requires a host with high HR efficiency and transformation efficiency [17]. Chase L. Beisel proposed and verified that the weakened CRISPR-Cas system could obtain more transformants during the transformation process, and improve genome editing efficiency to a certain extent [18].

In order to realize the efficient gene editing of the CRISPR/Cas9 system in wild-type *A. oryzae* and obtain positive transformants quickly, the preparation conditions of protoplasts were optimized, and the efficient transformation of *A. oryzae* was achieved. Based on the high transformation level, the gene editing efficiency of the CRISPR/Cas9 system in *A. oryzae* has also been improved. Furthermore, target transformants can be obtained accurately and quickly by screening for color or resistance. Finally, based on the optimized CRISPR/Cas9 system, multiple heterologous lipase gene copies can be integrated at different loci in one round of transformation.

## 2. Materials and Methods

### 2.1. Strain, Media, and Culture Conditions

*Aspergillus oryzae* RIB40 (ATCC42149) was used in this study to verify the transformation efficiency, CRISPR/Cas9 system editing efficiency, and heterologous protein expression. *A. oryzae* can be activated at 30 °C in potato dextrose agar (PDA) medium for 3–5 days, and the spore suspension can be obtained after elution and filtration with deionized water. The spore suspension was inoculated into adjusted Czapek Dox (CD) liquid medium (2% Glucose, 0.3% NaNO_3_, 0.1% K_2_HPO_4_, 0.05% KCl, 0.05% MgSO_4_·7H_2_O, 0.001% FeSO_4_·7H_2_O), and the fresh hyphae were obtained by shaking culture at 30 °C for 20 h, which were used for the preparation of protoplasts. The formula of fermentation medium for lipase includes 2% dextrin, 0.5% peptone, 0.1% yeast extract, 0.1% NaNO_3_, 0.05% KH_2_PO_4_, 0.05% MgSO_4_·7H_2_O, 0.001% FeSO_4_·7H_2_O. Hypertonic CD medium supplemented with 0.5 µg/mL pyrithiamine was used to detect positive transformants during transformation. For the growth of the *pyrG* mutant strain, hypertonic CD medium was supplemented with 0.2% uracil (Macklin, Shanghai, China) and 0.5% uridine (Macklin, Shanghai, China). Commonly used chemical reagents were purchased from Sinopharm (Shanghai, China).

### 2.2. Construction of Editing Plasmids and DNA Manipulation

Based on the pFC902 plasmid [16], the relevant elements of the CRISPR/Cas9 system were obtained by PCR, and the intermediate plasmid pC9sgR-model was constructed to modify the protospacer sequence. The linearized pPTR II vector and Cas9-sgRNA module were then assembled using GeneArt™ Gibson Assembly^®^ HiFi Cloning Kits (Thermo Fisher Scientific, Shanghai, China) to obtain editing plasmids. Using the In-Fusion^®^ HD Cloning Kit (Takara, Beijing, China), two sgRNAs can be tandemly linked to the Cas9-sgRNA module before the assembly of the dual gene editing plasmid. The primary primers used in this paper are listed in Appendix A. *E. coli* JM109 was used for DNA manipulation, and plasmids were extracted using SanPrep Column Plasmid Mini-Preps Kit (Sangon Biotech, Shanghai, China). Nucleotide sequencing analysis of target genes was performed commercially by Sangon Biotech (Shanghai, China).

### 2.3. Protoplast Preparation and Transformation

Liquid CD medium was inoculated with spore suspension obtained by elution from mature PDA plates, incubated with shaking at 30 °C for 16 h, and fresh *A. oryzae* hyphae were obtained after filtration and washing. The mycelium and the enzyme solution were mixed in proportion and shaken at 30 °C for 2 h. During this process, the composition of the complex enzyme, enzymatic hydrolysis conditions, and the growth state of mycelium were optimized. Three enzyme preparations were selected, including compound enzyme 1 (0.5% Glucanex and 0.05% Chitinase), compound enzyme 2 (1% cellulase, 1% helicase, 0.5% lysozyme, 0.5% Lywallzyme), and Yatalase (Takara, Beijing, China). Then, the protoplasts were collected and resuspended in STC buffer (1.2 M Sorbitol, 50 mM CaCl_2_, 50 mM Tris-HCl, pH 7.5). During transformation, 10 µg of DNA was added to 200 µL of protoplasts, and 80 µL of PEG4000 (*w*/*v* = 40%) was added simultaneously, mixed, and incubated on ice for 30 min. Then, 1.5 mL of PEG4000 was added to the above mixture and incubated at room temperature for another 30 min. Finally, the transformation solution was spread on the corresponding plates.

### 2.4. Editing Efficiency Statistics

When statistically calculating the random editing efficiency of morphological genes, such as *yA* and *wA*, the ratio *Rm* of a single colony with a color change in the transformation plate to the entire colony was first counted. Then, a certain number of single colonies with color changes and no changes were picked for colony PCR and sequencing verification, and the edited ratios were obtained as *Ec* and *En*, respectively [19]. The final editing efficiency is *e* = *Rm* * *Ec* + (1 − *Rm*) * *En*. For the statistics of the editing efficiency of *pyrG*, single colonies were picked from a transformation plate without selection pressure and cultured on a plate supplemented with 5-fluoroorotic acid (5-FOA) (Macklin, Shanghai, China). The proportion of the growing colonies to the total number of picked colonies was calculated as *Rp*, and then the growing colonies were analyzed. Colony PCR and sequencing validation obtained the editing ratio as *Ep*. The final editing efficiency is *e* = *Rp* * *Ep*. For the editing efficiency of conditional screening, we directly took *Ec* as the editing efficiency of morphological genes. For *pyrG*, we directly applied the transformation solution to the screening plate with 5-FOA, then randomly picked colonies for colony PCR and sequencing verification [19].

### 2.5. Colony PCR

A few fresh mycelia were picked into the lysis buffer (containing 0.5% Triton X-100, 1 mM EDTA, 50 mM NaOH), then lysed at 95 °C for 20 min. The supernatant obtained after centrifugation was used as a template for a standard PCR system.

### 2.6. Determination of the Growth Curve of A. oryzae

The fresh spore suspension was inoculated into CD medium and cultured with shaking at 30 °C. Three shake flasks were taken out every 12 h, and all the mycelia in the shake flasks were collected and dried [20].

### 2.7. Protein Expression and Validation

The spore suspension was inoculated into the fermentation medium, and cultured with shaking at 30 °C for 72 h. After centrifugation, the supernatant was collected, and the same volume of sample was controlled for SDS-PAGE analysis.

### 2.8. Determination of Lipase Enzyme Activity

Definition of enzyme activity: Under certain conditions, the amount of enzyme required to catalyze p-nitrophenol palmitate to 1 µmol of p-nitrophenol per unit of time is one unit of enzyme activity. Thus, 1.8 mL of 0.3% (*w*/*v*) p-nitrophenol palmitate (p-NPP) was mixed with 200 µL of enzyme solution, reacted at 40 °C for 10 min, and then 1 mL of 95% ethanol was added to terminate the reaction. After centrifugation, the absorbance at 410 nm of the reaction solution was measured [21].

### 2.9. Determination of α-Amylase Enzyme Activity

Definition of enzyme activity: Under certain conditions, the amount of enzyme that reacts to produce 1 mg of glucose per unit time is an enzyme activity unit U. A total of 100 μL enzyme solution, and 900 μL substrate were accurately reacted at 50 °C for 10 min. Then, 2 mL of 3,5-Dinitrosalicylic acid (DNS) was added, put in a boiling water bath for 10 min, and diluted to 25 mL after cooling. Absorbance was detected at 540 nm [22].

## 3. Results

### 3.1. Optimization of the Genetic Transformation

Screening markers ensure the rapid detection and purification of target transformants in gene transformation [23]. In order to establish an efficient genetic transformation process, the sensitivity of *A. oryzae* RIB40 to three antibiotics commonly used in *Aspergillus* was explored. This included hygromycin, bleomycin, and pyrithiamine. The wild-type *A. oryzae* RIB40 was very sensitive to pyrithiamine, but showed strong resistance to hygromycin. Bleomycin was not suitable as a resistance screen for *A. oryzae* RIB40 because *A. oryzae* RIB40 had strong tolerance within the range of commonly used concentrations of bleomycin (Appendix A). Considering the characteristics of heterokaryons, 0.5 µg/mL pyrithiamine with intense screening pressure was chosen for subsequent experiments.

The quality of protoplasts is a vital prerequisite for efficient transformation [24]. As recipient cells, protoplasts are not hindered by rigid cell walls and have the advantages of large population numbers and easy access to homozygous transformants. During the preparation of protoplasts, the growth state of mycelium and enzyme preparation directly affects the generation and release of protoplasts. In order to efficiently prepare a large number of excellent protoplasts, three enzyme preparations were selected, including compound enzyme 1 (0.5% Glucanex and 0.05% Chitinase), compound enzyme 2 (1% cellulase, 1% helicase, 0.5% lysozyme, 0.5% Lywallzyme), and Yatalase (Takara), for evaluation. The compound enzyme 2 had the best effect (Figure 1a). Then, the use conditions of compound enzyme 2 were optimized, such as temperature, pH, and enzymatic hydrolysis time (Figure 1b,d). At the same time, the culture time and addition amount of *A. oryzae* mycelium were also optimized (Figure 1e,f). After optimization, 5.2 × 10^6^ protoplasts/mL could be obtained by adding 10 mL of the compound enzyme to the 1.2g fresh mycelium, cultured for 24 h under 35 °C and pH 6.0 for two hours with shaking and enzymatic hydrolysis. The concentration of protoplasts was one-fold higher than before the conditions were optimized (Figure 1g). Subsequently, the transformation efficiency was evaluated based on the optimized protoplasts. The transformation experiments were performed in *A. oryzae* with the vector plasmid pPTR II, and we found that the transformation level of 150 colonies/10 μg DNA was achieved (Figure 1h). In addition, more transformants could be obtained by increasing the transformed DNA content.

### 3.2. Single-Gene Editing

As an endonuclease, the Cas9 protein is toxic to host strains, which may affect the expected growth of hyphae [25]. The growth curves of *A. oryzae* RIB40 containing vector plasmids pPTR II and pTR-Cas9 indicated that the Cas9 protein had only a slight inhibitory effect on the growth (Figure 2a). Morphological genes (*yA* and *wA*) and trophic genes (*pyrG*) were selected to validate the editing function and efficiency of the CRISPR/Cas9 system. Mutants of the *yA* (AO090011000755) gene encoding conidial laccase form yellow conidia, whereas *wA* (AO090102000545) mutants form white conidia due to the lack of the polyketide synthase required for conidia coloration. The *pyrG* (AO090011000868) gene encodes an orotidine-5’-phosphate decarboxylase, and deletion mutants of this gene are dystrophic for uridine and uracil. 5-FOA is a structural analogue of orotic acid that can enter the pyrimidine synthesis pathway. Under the action of a series of enzymes, such as orotidine-5′-monophosphate decarboxylase and thymidylate synthase, substances that are toxic to cells are produced, resulting in cell death. Editing experiments were performed on the three genes using the CRISPR/Cas9 system (Figure 2b and Appendix A). After statistics and analysis, the editing efficiency of random was up to 37.6%. Screening based on color or resistance increased editing efficiency to 100% (Figure 2c). The detected loss or insertion of fragments of different lengths in the target genes (Appendix A) resulted in changes in morphological and physiological properties (Figure 2d).

### 3.3. Dual Gene Editing

Metabolic engineering transformation of *A. oryzae* often requires editing or regulating multiple genes. It takes a lot of time and energy to go through multiple rounds of single-gene editing and requires continuous recycling of selectable markers. In order to improve the transformation efficiency, the double gene editing efficiency of the CRISPR/Cas9 system in *A. oryzae* was explored by targeting *yA* and *agdA* (AO090038000234), *yA* and *amyB*, *yA* and *ku70* (AO090011000936), respectively (Figure 3a). The morphological gene *yA* could be used as a screening marker to realize the rapid screening of multiple gene editing positive clones, indicating that *wA* with higher editing efficiency could achieve the same results or even better results. Both *agdA* and *amyB* are regulated by the transcription factor AmyR. The results indicated insertions or deletions of different lengths in both target genes (Appendix A). The efficiency of dual-gene editing decreased relative to single-gene editing, ranging from 18.5% to 19.8% (Figure 3b). Among them, in the double gene knockout of *yA* and *ku70*, the editing efficiency without adding donor DNA was significantly reduced because *ku70* is a functional protein in the NHEJ pathway. After knocking it out, the NHEJ repair pathway is destroyed. The double-strand breaks (DSB) caused by CRISPR/Cas9 are challenging to repair and thus cannot survive. It is worth noting that in the strains in which the morphological gene *yA* was successfully knocked out, the knockout rate of another gene was 100%, and the *yA* gene can be used as an indicator for the knockout of other genes, which also indicates that the transformation efficiency or Cas9 function might still be the limiting factors for genome editing. In addition, the α-amylase gene in *A. oryzae* has three copies, *amyA*, *amyB*, and *amyC*, and their sequences are highly identical [26,27]. Four strains with *adgA* and *amyB* mutations were selected and evaluated for the expression of α-amylase, respectively (Figure 3c). Knockout of the *agdA* increases the expression of α-amylase because of the abolition of competition for the transcription factor AmyR. Strain #5 obtained by one round of gene knockout completely lost the expression of α-amylase, indicating that the CRISPR/Cas9 system can effectively knock out multiple copies of genes simultaneously (Figure 3d). Strains #7 and #8 had point mutations; they might not affect enzyme activity. However, strain #6 with frameshift mutation did not cause significant changes in amylase activity, which requires further experiments to explore and explain.

### 3.4. CRISPR/Cas9-Mediated Multiplexed Site Integration in A. oryzae for Heterologous Lipase Expression

The heterologous lipase gene (*TLL*) from *Thermomyces lanuginosa* was integrated into the *amyB* locus to verify the efficient genome editing of the CRISPR/Cas9 system in *A. oryzae*. The three genes *amyA*, *amyB*, and *amyC* were consistent in the first 2658 bp, including the promoter, while the *amyA* gene lacked the terminator in the subsequent sequence [28]. Therefore, the CRISPR/Cas9 system could target and cut three genes simultaneously, but donor DNA (PamyB-lipase-TamyB) could only target and repair the *amyB* and *amyC* genes. The pTR-C9sgR-yA-amyB editing plasmid and the donor DNA containing the lipase expression cassette were simultaneously transformed (Figure 4a and Appendix A). Under the indication of the morphological gene *yA*, the mutant strains whose lipase gene was successfully integrated into the α-amylase locus could be selected accurately and quickly (Figure 4b). Subsequently, the situation of the integration site of the *TLL* was explored (Figure 4b). Of the six transformants picked, three transformants achieved double-site integration. In order to verify the expression of TLL in these transformants, fermentation experiments were conducted. Transformants integrated only at the *amyC* site had the lowest TLL expression levels and some alpha-amylase expression. However, the transformants integrated into the *amyB* site alone did not express α-amylase, and the expression level of TLL was also increased. The *amyB* locus is better than the *amyC* locus for expressing heterologous genes, but the mechanism needs to be further explored and verified. Transformants with double-copy *TLL* had higher expression levels, which verified that increasing the copy number of genes in *A. oryzae* could effectively increase the expression level (Figure 4c).

## 4. Discussion

*A. oryzae* has been widely used to efficiently produce organic acids, industrial enzymes, and secondary metabolites, due to its efficient protein expression ability and post-translational modification functions [29]. However, the problem of low expression of heterologous proteins is a bottleneck that cannot be ignored in the development and application of *A. oryzae* expression systems, and the complex and multi-level regulatory mechanisms are still unclear [30]. Therefore, establishing efficient genome editing tools is conducive to the faster and better development of *A. oryzae* cell factories and the analysis of their high-efficiency expression mechanisms.

Establishing an efficient genetic transformation system plays a fundamental role. According to the main components of the cell wall of *A. oryzae*, the types of mixed enzymes used in the preparation of protoplasts were optimized. Coupled with the optimization of the relevant conditions for protoplast preparation, good and high-quality protoplasts were finally obtained.

The CRISPR/Cas9 editing system has an outstanding application prospect in *A. oryzae*. Currently, the editing efficiency of the CRISPR/Cas9 system in wild-type *A. oryzae* is 50–100% under screening pressure conditions [15]. However, in the construction of the *A. oryzae* expression system, the genes that need to be edited often do not have unique screening pressures. The acquisition of positive transformants still takes a lot of time and effort. In our study, the editing efficiency of CRISPR/Cas9 system in single gene and double gene in *A. oryzae* was 17.5-37.6% under the condition of no screening pressure. Likewise, the editing efficiency can be increased to 100% by color or resistance-aided screening. Notably, the morphological gene *yA* can be used as an indicator marker for other gene editing, which is of great help for the rapid and precise acquisition of positive transformants in the gene editing of *A. oryzae*.

Achieving simultaneous editing of multiple genes is beneficial to improve the use efficiency of selection markers and analyze the cooperation of many related genes [31]. Currently, the CRISPR/Cas system has progressed in multi-gene editing in filamentous fungi. Tian et al. used the CRISPR/Cas9 system to edit the double, triple, and quadruple genes in filamentous fungi *Myceliophthora* with efficiencies of ~60%, ~30%, and ~20%, respectively [32]. In addition, based on the optimized CRISPR/Cas9 system, Liu et al. achieved efficient multi-gene editing in *Aspergillus niger*, and the editing efficiencies of double, triple, and quadruple genes were 70.91%, 50%, and 22.41%, respectively [19]. In this study, when *yA* and three copies of the amylase gene were edited, it could theoretically be considered simultaneous editing of four genes, and the editing efficiency of the knockout was 4.9%. The subsequent *TLL* integration expression experiment can also be considered as the simultaneous editing of three genes, and the editing efficiency of knock-in is 9.9%. These results indicate that it is feasible to achieve simultaneous knockout of multiple genes or integrated expression of multiple genes in *A. oryzae*, although the current efficiency is still relatively low. Therefore, it is necessary to improve the editing efficiency of multiple genes through other measures [33]. Moreover, using appropriate endogenous tRNAs to achieve efficient maturation of multiple sgRNAs could also improve editing efficiency [19].

In conclusion, a feasible CRISPR/Cas-mediated multiplexed gene editing tool was developed in *A. oryzae,* and the genetic editing efficiency was further improved after optimization. However, achieving rapid editing of multiple genes requires the host to have multiple copies of the target gene. The morphological gene *yA* can be used as a promising selection marker for the rapid and accurate acquisition of positive transformants during gene editing of *A. oryzae*. Finally, the CRISPR/Cas9 system achieved multi-copy integration of the same gene or simultaneous integration of different genes in *A. oryzae*, which will also provide easy-to-use tools for constructing filamentous fungi cell factories.

## Figures and Tables

**Figure 1 jof-09-00109-f001:**
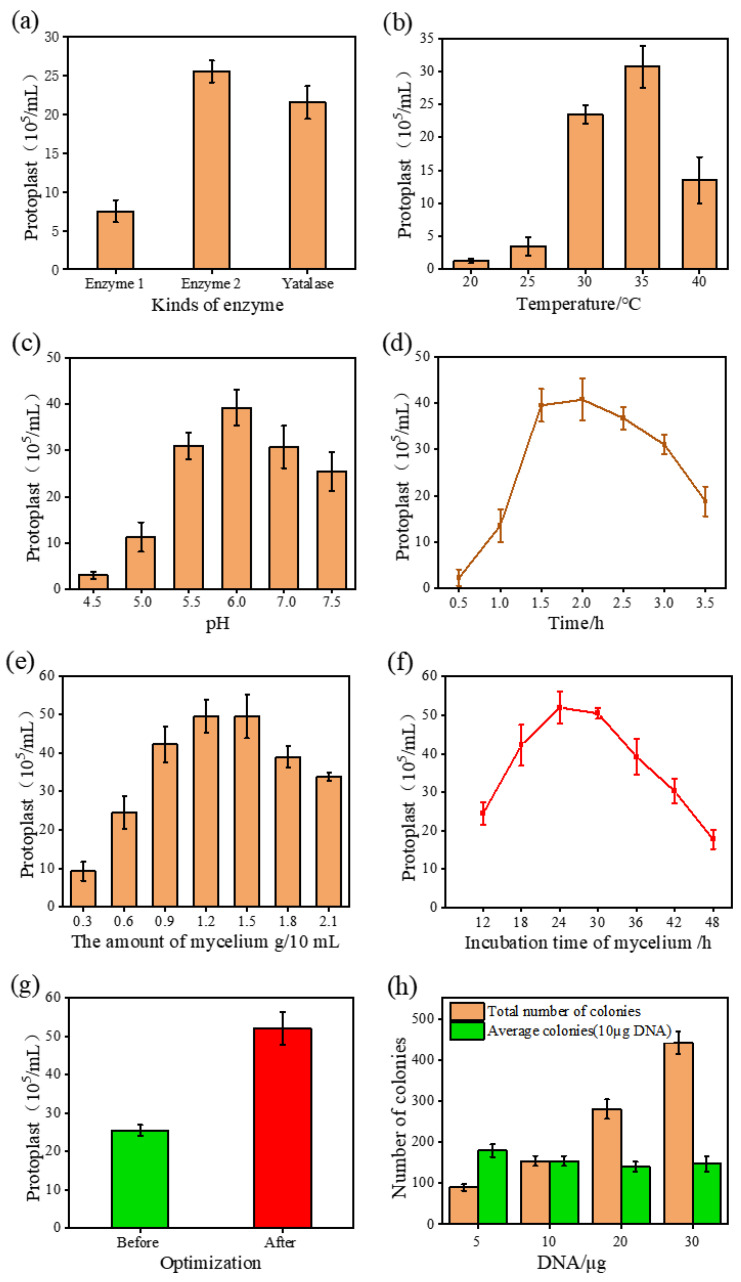
Optimization of conditions for protoplast preparation. (**a**) Comparison of different complex enzymes. (**b**) Effect of temperature on protoplast preparation. (**c**) Effect of pH on protoplast preparation. (**d**) Effect of hydrolysis time on protoplast preparation. (**e**) Effect of mycelium amount on protoplast preparation. (**f**) Effect of mycelial culture time on protoplast preparation. (**g**) The effect of optimization of protoplast preparation. (**h**) Transformation efficiency of different DNA contents after optimization.

**Figure 2 jof-09-00109-f002:**
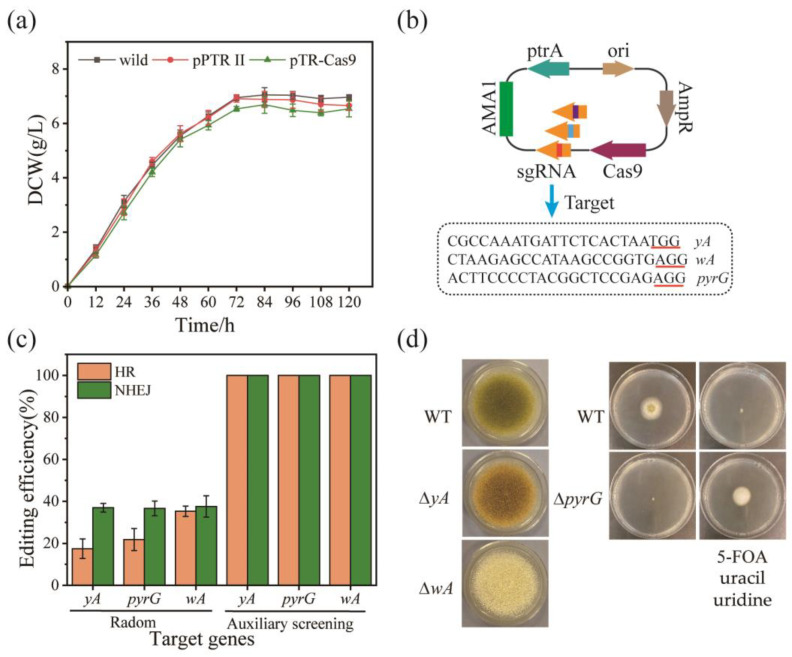
CRISPR/Cas9 mediated single-gene editing in *A. oryzae*. (**a**) The effect of Cas9 protein on the growth of *A. oryzae*. (**b**) Schematic diagram of editing plasmids and their target sequences. The bases underlined in red represent the protospacer adjacent motif (PAM) sequence. (**c**) The editing efficiency of related genes by CRISPR/Cas9 system. In experiments based on HR repair, donor DNA is the homology arm that deletes large fragments of CDS. n = 5. (**d**) Phenotypes of *yA*, *wA*, and *pyrG* mutants were generated from the wild strain. Wild-type *A. oryzae* cannot grow normally on the plate in the presence of 5-FOA, while the *pyrG* mutant strain can grow normally when additional uracil and uridine are needed. Mutants of the *yA* gene encoding conidial laccase form yellow conidia, whereas *wA* mutants form white conidia due to the lack of the polyketide synthase required for conidia coloration.

**Figure 3 jof-09-00109-f003:**
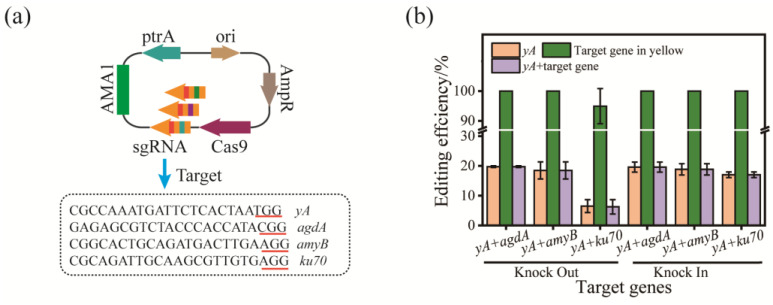
CRISPR/Cas9 mediated dual gene editing in *A. oryzae*. (**a**) Schematic diagram of editing plasmids and their target sequences. The bases underlined in red represent the PAM sequence. (**b**) The editing efficiency of double genes by CRISPR/Cas9 system. The “target gene in yellow” represented the editing of another target gene (*agdA*, *amyB*, or *ku70*) in the selected yellow colonies (∆*yA*). n = 5. (**c**) Sequence profile of selected mutant strains. Green, red, orange, blue, and purple letters represent the protospacer, PAM, deletion, insertion, and mutated sequences, respectively. * means the gene contains mutations. (**d**) Enzyme activity and protein expression level of α-amylase in mutant strains.

**Figure 4 jof-09-00109-f004:**
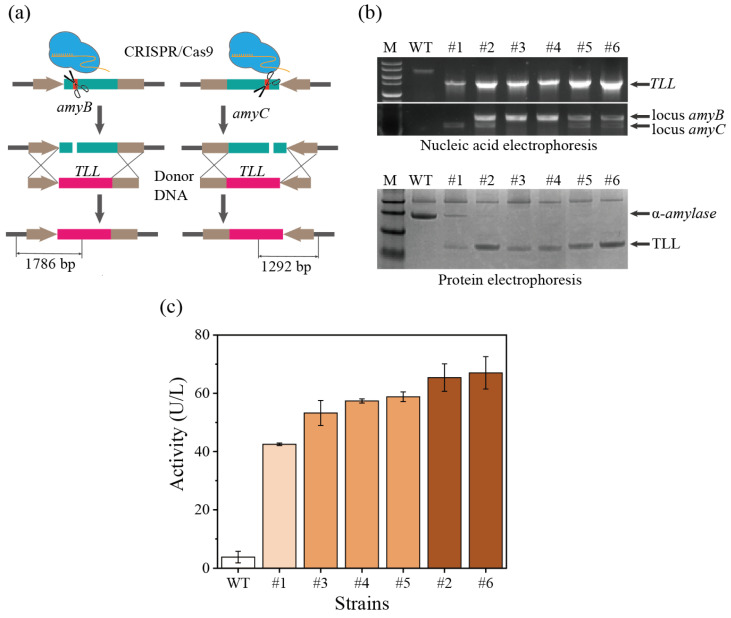
Integrative expression of *TLL* in *A. oryzae*. (**a**) Schematic diagram of the integrated expression of *TLL* in *A. oryzae*. The indicated sequence lengths were used to verify different integration sites using primers Locus-amyB-F, Locus-amyC-F, and Locus-TLL-R, respectively. (**b**) Different electrophoretic validation of *TLL* integrated expression. Nucleic acid electrophoresis showed that the *TLL* gene was successfully integrated into *amyB* or/and *amyC* sites, and protein electrophoresis showed that the integration of multiple copies was beneficial to improve the expression level of TLL. (**c**) Lipase activity after integrated expression of *TLL* in *A. oryzae*. n = 3. #1–6 represented the selected positive mutants.

## Data Availability

Not applicable.

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
