# Peer review of "CRISPR/Cas9-Mediated Multiplexed Genome Editing in Aspergillus oryzae"

_jof, 2023, doi:10.3390/jof9010109_

Round 1

Reviewer 1 Report

The manuscript described the high-efficiency transformation of A. oryzae by optimizing the preparation conditions of protoplasts and the random genome editing method by using CRISPR/Cas9 system in A. oryzae for single and double genes. The morphological gene yA could be used as a helpful selection marker for genome editing. It was demonstrated that the heterologous lipase gene (TLL) from Thermomyces lanuginosa was integrated into the amyB locus to verify the efficiency of genome editing of the CRISPR/Cas9 system in A. oryzae. The results concluded that the CRISPR/Cas9 system achieved multi-copy integration of the same gene or simultaneous integration of different genes in A. oryzae. The merit of this study is the establishment of a protoplast preparation method for efficiently transforming A. oryzae; however, the results and the descriptions of the genomic editing studies were not sufficient for recommendation to be published in the Journal of Fungi. The reasons are listed below.

Major:

1. It was described that bleomycin has no significant inhibitory effect on the growth of A. oryzae RIB40 (section 3.1). How to define significant or nonsignificant? The description was inappropriate because the use of bleomycin may be significant in a higher concentration.

2. Fig. 1h showed the transformation efficiency of different DNA contents after optimization but was inappropriate. It was described that the transformation level of 150 colonies/10 μg DNA could be achieved by transforming more than 5 μg of DNA (section 3.1). Both the figure and the description were not easy to understand. The bars in green, colonies/10 μg, seem unreasonable in Fig. 1h.

3. It was described that the editing efficiency of random was up to 46.8% (section 3.2). However, none of them exceeded 40% (Fig 2c).

4. It was described that the detected loss or insertion of fragments of different lengths in the target genes resulted in changes in morphological and physiological properties (section 3.2). The changes in morphological and physiological properties should be clearly described. In addition, the results of the plates added with 5-Fluoroorotic Acid (5-FOA) should be described (Fig 2d).

5. The reasons for using yA for further dual gene editing should be explained because it showed lesser editing efficiency than the other two (section 3.2).

6. It was described that the efficiency of dual-gene editing decreased relative to single-gene editing, ranging from 18.5% to 19.8%. The description was not accurate because that of double gene knockout of yA and Ku70 was lower (Fig 3b). In addition, the definition of the green bars in Fig 3b should be explained. What is the definition of the target gene in yellow?

7. It was described that strain #1 obtained by one round of gene knockout completely lost the expression of α-amylase. However, it showed a relatively higher α-amylase activity in Fig 3d. In addition, Fig 3d needs control to show equal amounts of proteins were loaded.

8. It was described that the activity of α-amylase was detected by the DNS method in the legend of Fig 3d. The method should be described in the Materials and Methods as that of lipase activity.

9. It was described that several other mutants of amyB did not cause significant changes in amylase expression, presumably because the presence of introns terminated the frameshift mutation. The speculation should provide shreds of evidence (section 3.3).

10. It was described that the gene sequences of amyB and amyC are highly identical, so the lipase expression cassette may target both the amyB and amyC locus (section 3.4). Actually, it was described that the α-amylase gene in A. oryzae has three copies, amyA, amyB, and amyC, and their sequences are highly identical (section 3.3). Why the lipase expression cassette did not target amyA locus?

11. It was described that in this study, when yA and three copies of the amylase gene were edited, it could theoretically be considered simultaneous editing of four genes, and the editing efficiency of the knockout was 4.9%. The subsequent TLL integration expression experiment can also be considered as the simultaneous editing of three genes, and the editing efficiency of knockin is 9.9% (section 4). The sources of the calculation of the percentage should be provided. In addition, the descriptions of theoretically considered simultaneous editing of four or three genes were inappropriate. The descriptions of CRISPR/Cas9 system achieved multi-copy integration of the same gene, or simultaneous integration of different genes in A. oryzae (section 4) were excessively interpretated according to the current data.

Reviewer 2 Report

This is a manuscript review for the article ''CRISPR/Cas9-mediated multiplexed genome editing in Aspergillus oryzae''

The work brings to light new aspects of Aspergillus oryzae. The issue below can improve the manuscript and help others reproduce the work

Methods: Major revision required- Not all sections showed where chemicals used were sourced. This is critical to enable others to reproduce the work. Did the authors make up the Czapek Dox medium? Name manufacturers and country of all chemicals used. Where is Sangon Biotech? What country?

Lines 113-114: Revise English

Lines 113-126: Provide the reference  for  editing efficiency statistics

Section 2.6: Revise and state more explicitly how the growth curve was carried out with at least a relevant reference.

Lines 158-162: Compound enzymes 1 and 2 were not mentioned in the methods section. This needs to be included in the method section. The reference for the optimisation process needs to be cited. If the optimisation method has no reference, the authors should give an explanation in the discussion section and state the reason for optimising the way they did.

Lines  146-147: revise English. Put a full stop after 'explored.' Remove 'including' and replace with 'This included'

Figure 1:

Optimization of conditions for protoplast preparation namely Effect of temperature on protoplast preparation. (c) Effect of pH on protoplast preparation. (d) Effect of hydrolysis time on protoplast preparation. (e) Effect of mycelium amount on protoplast preparation. (f) Effect of mycelial culture time on protoplast preparation. (g) The effect of optimization of protoplast preparation is not well described in the method section. This should be done.

Lines 179 and 202: Single and dual gene editing are not described in the methods section

Lines 283-284:Authors state '' Likewise, the editing efficiency can be increased to 100% by color or  resistance-aided screening''. This is a supposition. No such result was obtained., If it is a prediction, then the English needs to be revised. Words like 'may' should be used with appropriate references.

Line 307: Considering that the efficiency of 4.9% was obtained when yA was used, why is it considered a 'powerful' selection marker when earlier reports as authors have indicated, obtained a much higher of 70% efficiency?

The limitation of the methods proposed should be included in the conclusions.

Round 2

Reviewer 1 Report

The manuscript described the high-efficiency transformation of A. oryzae by optimizing the preparation conditions of protoplasts and the random genome editing method by using CRISPR/Cas9 system in A. oryzae for single and double genes. The morphological gene yA could be used as a helpful selection marker for genome editing. It was demonstrated that the heterologous lipase gene (TLL) from Thermomyces lanuginosa was integrated into the amyB locus to verify the genome editing efficiency of the CRISPR/Cas9 system in A. oryzae. The results concluded that the CRISPR/Cas9 system achieved multi-copy integration of the same gene or simultaneous integration of different genes in A. oryzae. The revised manuscript addressed many questions and made clear explanations. Now it is suggested to be published in the Journal of Fungi after modifications. Below are the suggestions for further improvement.

1. The gene name should be italicized. For example, four genes in Fig 3a. In addition, the gene name begins with a lowercase letter. For example, amyB gene in Fig 3b.

2. The methods used in this study should provide references. For example, the lipase enzyme activity.

3. The authors didn’t explain why strain #1 obtained by one round of gene knockout completely lost the expression of α-amylase. According to the authors’ reply and Fig 3d, it is strain #5 but not strain #1. The authors’ explanation for strains #5, #6, #7, and #8 can be added to the text. In addition, the method for protein expression should be described in the Materials and Methods, and explain that an equal volume of samples was loaded.

4. The authors’ reply for using yA for further dual gene editing should be added to the text.

5. The authors’ reply for the α-amylase genes, amyA, amyB, and amyC, in A. oryzae should be added to the text.

6. The authors’ reply for the sources of the calculation of the percentage can be added to the text. In addition, the authors’ reply for CRISPR/Cas9 system achieved multi-copy integration of the same gene, or simultaneous integration of different genes in A. oryzae can also be added.

Reviewer 2 Report

Manuscript has improved

Author Response

Thank you so much for your valuable suggestions. At the same time, we are also happy for your satisfaction with our modification.